# The Effect of SARS-CoV-2 Vaccination on B-Cell Phenotype in Systemic Sclerosis Patients

**DOI:** 10.3390/jpm12091420

**Published:** 2022-08-31

**Authors:** Chiara Pellicano, Amalia Colalillo, Valerio Basile, Mariapaola Marino, Umberto Basile, Francesca La Gualana, Ivano Mezzaroma, Marcella Visentini, Edoardo Rosato

**Affiliations:** 1Department of Translational and Precision Medicine, Sapienza University of Rome, 00185 Rome, Italy; 2Clinical Pathology Unit and Cancer Biobank, Department of Research and Advanced Technologies, IRCCS Regina Elena National Cancer Institute, 00144 Rome, Italy; 3Department of Translational Medicine and Surgery, Section of General Pathology, “A. Gemelli” IRCCS, Catholic University of the Sacred Heart, 00168 Rome, Italy; 4Department of Laboratory and Infectious Disease Sciences, “A. Gemelli” IRCCS, Catholic University of the Sacred Heart, 00168 Rome, Italy

**Keywords:** B cells, COVID-19, humoral response, mRNA vaccine, systemic sclerosis

## Abstract

Objective: to assess the influence of SARS-CoV-2 mRNA vaccine on B-cell phenotypes in systemic sclerosis (SSc). Methods: peripheral blood B-cell subpopulations were evaluated before (t1) and 3 months (t3) after the second dose of vaccine in 28 SSc patients. Peripheral blood B-cell subpopulations were evaluated in 21 healthy controls (HCs) only at t1. Anti-spike IgG levels were evaluated at t3 in both cohorts. Results: SSc patients presented higher naive, double-negative, and CD21^low^ B cells compared to HCs. IgM-memory and switched-memory B cells were lower in SSc patients than HCs. No differences in anti-spike IgG levels after vaccination were observed between SSc patients and HCs. Anti-spike IgG levels after vaccination were lower in SSc patients with increased CD21^low^ B cells at baseline compared to SSc patients with normal CD21^low^ B cells. A positive correlation was found between IgG levels and naive B cells. A negative linear correlation was shown between IgG levels and IgM-memory, switched-memory, double-negative, and CD21^low^ B cells. Conclusions: SARS-CoV-2 mRNA vaccine response is normal in SSc patients not undergoing immunosuppressive therapy. The normal number of naive B cells is a positive marker of antibody response. The increased percentage of CD21^low^ B cells represents a negative marker of antibody response.

## 1. Introduction

The use and safety of vaccines in autoimmune diseases has always been debated [1]. However, most investigated vaccines are immunogenic, although some are less effective in patients receiving immunosuppressive therapies, and vaccination did not seem to lead to an increase in activity of autoimmune disease [2].

The recent outbreak of the coronavirus disease 2019 (COVID-19) pandemic led the World Health Organization (WHO) to recognize the need for a safe and efficient vaccine to manage the spread of the severe acute respiratory syndrome coronavirus 2 (SARS-CoV-2) infection worldwide [3].

SARS-CoV-2 mRNA vaccines elicit potent SARS-CoV-2-specific germinal center (GC) B-cell responses and GC B cells strongly correlated with the production of neutralizing antibodies (nAbs). The mRNAs encoding SARS-CoV-2 antigens are translated into proteins directly in the host and this could lead to more prolonged antigen availability along with continuous antigen presentation via MHC II [4].

It is reported that patients with autoimmune diseases develop a significant humoral response following the administration of the second dose of mRNA vaccine for SARS-CoV-2, and there seems to be no impact on disease activity [5]. Studies to date have focused on the evaluation of the vaccine response by measuring neutralizing antibodies against the spike protein and on the persistence of an adequate antibody response by means of serial assessments of the antibody titer [6,7,8,9]. B lymphocytes in systemic sclerosis (SSc) patients show a constitutive activation profile, as demonstrated by hypergammaglobulinemia and the production of serum autoantibodies [10]. There is an increased frequency of naive B cells and of CD21^low^ B cells, a particular population of functionally anergic and exhausted B cells which possibly play the role of antigen-presenting cells (APCs). In SSc patients, increased CD21^low^ B cells are associated with pulmonary and renal vascular involvement and predict the development of digital ulcers [11,12]. There are no data regarding the influence of the mRNA vaccine against SARS-CoV-2 on B-cell phenotypes in SSc patients.

The aim of the study was to assess the influence of the SARS-CoV-2 mRNA vaccine on B-cell phenotypes in SSc patients.

## 2. Patients and Methods

### 2.1. Subjects

Twenty-eight SSc patients, fulfilling the American College of Rheumatology/European League Against Rheumatism Collaborative Criteria for SSc [13], and 21 healthy controls (HCs), matched for sex and age, were enrolled in this study. All study participants were administered the two-dose regimen of BNT162b2 mRNA vaccine (Pfizer-BioNTech Pfizer Inc., 235 East 42nd Street, New York, NY 10017, 1-800-879-3477 USA and BioNTech SE, An der Goldgrube 12, 55131 Mainz, Deutschland), 30 mcg per dose, by intramuscular injection 3 weeks apart, as indicated by the national guidelines. SSc patients continued all medications according to international recommendations [14,15].

Exclusion criteria were a reported previous SARS-CoV-2 infection, patients receiving only one dose of vaccine, and <18 years, or previous or concomitant therapy with rituximab or mycophenolate mofetil (MMF). The 28 patients included in the study did not receive any immunosuppressive treatment or disease-modifying anti-rheumatic drugs (DMARDs) in the last 6 months before vaccination and during the period of vaccine administration and observation, except 4 patients who received corticosteroids at an equivalent dose of prednisone < 10 mg/day.

The subjects’ written consent was obtained and the study was conducted according to the Declaration of Helsinki. The study was approved by the ethics committee of Sapienza University (IRB n. 0486).

### 2.2. Laboratory Assays

Peripheral venous blood samples were collected before injection of the first dose of vaccine (t1) and 3 months after the second vaccine dose (t3). B-cell immunophenotype was assessed both at t1 and t3 for SSc patients and only at t1 for HCs, conversely anti-spike IgG serum levels were evaluated only at t3 in both cohorts. 

Flow cytometry analysis was performed on a BD FACSCalibur system (Becton Dickinson, Mountain View, CA, USA) and data files were acquired and analyzed using CELLQuest 3.3 (Becton Dickinson, Mountain View, CA, USA) and FlowJo (TreeStar, Ashland, OR, USA) software; the gate strategy was the same as our previous study [11]. The gating strategy of a representative patient and HC is shown in Figure 1. 

Serum samples were tested with LIAISON® SARS-CoV-2 TrimericS IgG assay (DiaSorin, Saluggia, Italy) to measure anti-spike IgG levels. Results are provided in binding antibody units (BAU) with a quantification range of 4.81–2080 BAU/mL, and values ≥ 33.8 BAU/mL were considered positive.

### 2.3. Clinical Assessment of SSc Patients

In SSc patients, modified Rodnan skin score (mRSS), disease subset (limited cutaneous SSc, lcSSc, or diffuse cutaneous SSc or dcSSc), disease duration, disease activity index (DAI), and disease severity scale (DSS) were assessed [16]. Nailfold videocapillaroscopy (NVC) was performed at the level of the distal phalanx of the second, third, and fourth fingers of both hands using a videocapillaroscope equipped with a 500x magnification lens (Pinnacle Studio Version 8 software) and the capillaroscopic images have been classified in the patterns early, active, and late [17].

### 2.4. Statistical Analysis

SPSS version 26.0 software (IBM Corp. Released 2019. IBM SPSS Statistics for Windows, Version 26.0. Armonk, NY, USA: IBM Corp) was used for statistical analysis. After evaluation of normality, continuous variables were expressed as median and interquartile range (IQR). Mann–Whitney’s *t*-test was used to evaluate differences between groups. Bonferroni’s corrections were applied in case of multiple comparisons. The Fisher exact test was used to evaluate differences between categorical variables. The Spearman correlation test was used for bivariate correlations. The cutoff values for increased B-cell subset distribution were chosen as media +2 standard deviation of the B-cell subsets percentage of HCs as follows: increased naive ≥ 80.4%, increased IgM memory ≥ 43.8%, increased switched memory ≥ 36.2%, increased double negative ≥ 6.5%, and increased CD21^low^ ≥ 9%. *p*-value < 0.05 was considered significant.

## 3. Results

Demographic and clinical features of SSc patients and HCs are shown in Table 1.

### 3.1. B-Cell Subset Distribution at t1 and Serum Anti-Spike IgG Levels in SSc Patients and HCs

No differences in total B-cell percentage were observed between SSc patients and HCs [7.53% (IQR 5.87–11.5) vs. 7.6% (IQR 6.86–10.2), *p* > 0.05]. Naive B-cell percentage was higher in SSc patients than HCs [70.55% (IQR 47.15–80.2) vs. 55.4% (IQR 44.5–62.7), *p <* 0.05]. IgM memory [8.57% (IQR 4.62–19.15) vs. 21.3% (IQR 15.8–29.8), *p <* 0.001] and switched memory [14.15% (IQR 8.5–20.2) vs. 19.5% (IQR 16.1–21.9), *p <* 0.05] were significantly lower in SSc patients than HCs. Double-negative B cells [5.16% (IQR 3.78–7.76) vs. 3.96% (IQR 3.03–5.1), *p <* 0.05] and CD21^low^ B cells [5.59% (IQR 3.84–9.13) vs. 3.65% (IQR 3.11–4.69), *p <* 0.05] were significantly higher in SSc patients than HCs.

No differences in serum IgG levels were observed between SSc patients and HCs [635.5 BAU/mL (IQR 285–1135) vs. 358.8 BAU/mL (IQR 177.06–1021.8), *p >* 0.05] at 3 months after vaccination.

These findings are summarized in Table 2.

### 3.2. B-Cell Subset Distribution 3 Months after Vaccination in SSc Patients

The percentages of B cells and of double-negative B cells were unchanged from baseline. Naive B cells were reduced, whilst IgM-memory, switched-memory, and CD21^low^ B cells were increased. Table 3 and Figure 2 summarize these findings. 

Stratifying SSc patients according to antibody positivity (anti-topoisomerase I and anti-RNA polymerase III vs. anti-centromere and ANA), we did not find any statistically significant differences in antibody response nor in the distribution of B-cell subpopulation both at t1 and t3. Moreover, stratifying SSc patients according to interstitial lung disease (ILD), we did not find any statistically significant differences in antibody response or in the distribution of B-cell subpopulations at t1 nor t3. These data are reported in Table 4.

### 3.3. Anti-Spike IgG Serum Levels 3 Months after Vaccination and Their Correlation with B-Cell Subpopulations at Baseline

Serum IgG levels after vaccination were higher in SSc patients with increased naive B cells at baseline (*n =* 8, 28.6%) than SSc patients with normal naive B cells [1350 BAU/mL (IQR 756–1480) vs. 470.5 BAU/mL (IQR 259–937); *p <* 0.01]. Serum IgG levels after vaccination were significantly lower in SSc patients with increased CD21^low^ B cells at baseline (*n =* 6, 21.4%) compared to SSc patients with normal CD21^low^ B cells [212 BAU/mL (IQR 129–304) vs. 756 BAU/mL (IQR 439–1370); *p <* 0.01]. No differences in serum IgG levels were observed in SSc patients with normal or increased (*n =* 20, 71.4%) double-negative B cells [750 BAU/mL (IQR 439–1330) vs. 337.5 BAU/mL (IQR 210–946), *p >* 0.05].

The baseline percentage of B cells did not correlate with serum IgG level either in SSc patients (r = 0.242; *p >* 0.05) nor in HCs (r = 0.156, *p >* 0.05). Among SSc patients a significant positive linear correlation was found between serum IgG levels and percentage of naive B cells (r = 0.472; *p <* 0.001) (Figure 3 panel A) and a significant negative linear correlation was observed between serum IgG levels and percentage of IgM-memory (r = −0.327; *p <* 0.05), switched-memory (r = −0.478; *p <* 0.001), double-negative (r = −0.310; *p <* 0.05), and CD21^low^ B cells (r = −0.389; *p <* 0.01) (Figure 3 panel B). We did not find any correlation between B-cell subpopulations and antibody titers in HCs: naive B cells (r = −0.014, *p >* 0.05), memory B cells (r = −0.077, *p >* 0.05), switched-memory B cells (r = −0.043, *p >* 0.05), double-negative B cells (r = −0.063, *p >* 0.05), and CD21^low^ B cells (r = 0.272, *p >* 0.05).

## 4. Discussion

The results of this study confirm that naive B cells, double-negative B cells and CD21^low^ B cells are increased in SSc patients compared to HCs. At 3 months after vaccination, no differences in serum IgG levels were observed between SSc patients and HCs. IgM-memory, switched-memory, and CD21^low^ B cells increased at 3 months after vaccination; conversely, naive B cells were reduced. Anti-spike IgG serum levels 3 months after vaccination were higher in SSc patients with normal naive B cells at baseline and lower in SSc patients with increased CD21^low^ B cells at baseline. At 3 months after vaccination, a significant positive linear correlation was found between anti-spike IgG serum levels and percentage of naive B cells; conversely, a significant negative linear correlation was observed between serum anti-spike IgG serum levels and percentage of CD21^low^ B cells.

In a previous study, we found that CD21^low^ B cells were increased in SSc patients and were a marker of visceral vascular involvement [11]. The present study confirms the increased levels of CD21^low^ B cells in SSc patients.

The serum anti-spike IgG levels 3 months after second vaccination dose were similar between SSc patients and HCs. Several studies have investigated the efficacy of immunosuppressive therapy on SARS-CoV-2 vaccination in patients with autoimmune inflammatory diseases, showing that antibody responses are delayed and reduced compared to HCs [6,7,8,9,18]. However, it remains to be clarified whether the antibody response deficit is due to disease itself or to immunosuppression. Ferri et al. demonstrated that the immunogenicity of mRNA COVID-19 vaccines was reduced in autoimmune systemic diseases (ASD). Increased prevalence of non-response to vaccine was observed in patients with ASD-related interstitial lung disease, and in those treated with glucocorticoids, MMF, or rituximab [9]. Although in the present study there are no differences in response to vaccination or B-cell populations between patients with SSc-ILD and patients without ILD, this may be due to the small sample size that prevents further analysis. After full vaccination with inactivated SARS-CoV-2 vaccine, Sampaio-Barros et al. demonstrated that anti-SARS-CoV-2-IgG frequency and neutralizing antibody positivity were moderate, although lower than controls. Vaccine antibody response is not influenced by disease subtype and is greatly affected by MMF [18]. Delayed antibody responses to the SARS-CoV-2 vaccine may suggest an effect of immune-modulatory treatments. Simon et al. found that patients with autoimmune inflammatory diseases without treatment also had lower antibody responses than controls, so the delayed antibody responses seem to be a disease-related rather than a treatment-related effect [19]. In our previous study, we showed that the seropositivity conversion was similar 1 month after the second injection but significantly lower 3 months after the completion of the vaccination cycle in SSc patients compared to HCs. It is noteworthy that the antibody titer was lower 1 month after the second injection and similar 3 months after the completion of the vaccination cycle between SSc patients and HCs. Interestingly, in a sub-analysis comparing only non-immunosuppressed SSc patients with HCs, the seropositivity rate was similar both 1 month and 3 months after the completion of the vaccination cycle, and SSc patients without immunosuppressive therapy had statistically significant lower serum IgG levels than HCs at 1 month but not 3 months after the completion of the vaccination cycle [20]. In our study group, the vaccine antibody response was normal probably because immunosuppressive treatment was an exclusion criterion. We can suppose that vaccine antibody response is normal in SSc patients.

SARS-CoV-2 vaccines are able to induce specific cellular and humoral responses generating immune memory, as activation of B cells plays a key role in the effectiveness of the response to vaccination [4]. Morales-Núñez et al. analyzed the changes induced in the different cellular subpopulations of B cells after vaccination with BNT162b2 in people with and without a history of COVID-19, comparing the results with a group of individuals not vaccinated against SARS-CoV-2. They found that the BNT162b2 vaccine induces changes in B-cell subpopulations, especially generating plasma cells and producing neutralizing antibodies against SARS-CoV-2. Moreover, previous infection with SARS-CoV-2 did not significantly alter the dynamics of these subpopulations but induced more rapid and optimal antibody production [21]. In this study, we found a reduction of naive B cells and an increase in IgM-memory, switched-memory, and CD21^low^ B cells in SSc patients after vaccination. The serum anti-spike IgG levels after vaccination showed a positive correlation with baseline percentage of naive B cells and negative correlation with baseline percentage of IgM-memory, switched-memory, double-negative, and CD21^low^ B cells in SSc patients but not in HCs. This difference is probably due to the alteration of B-cell homeostasis that characterizes SSc patients [22]. Schulz et al. found a strong association between the number of circulating naive B cells and serum anti-spike IgG in a cohort of hematologic immunocompromised patients; moreover, naive B-cell count was the only independent predictive factor for achieving antibody titers comparable with HCs in multivariable analysis [23]. 

CD21^low^ B cells accumulate in different disease settings characterized by chronic (auto)antigen stimulation and show features of exhausted and apoptosis-prone cells [11,24]. Their origin is still elusive, but their formation seems to rely on TLR9 or TLR7 signals in the context of Th1 cytokines, particularly IFN-γ [25]. In health and autoimmune diseases CD21^low^ B cells seem to have a role as (auto)APCs, potentially contributing to the autoimmune reaction [26]. Therefore, in SSc patients with higher percentages of CD21^low^ B cells, anti-spike IgG response might be reduced as these cells accumulate as activated, mostly autoreactive and exhausted cells in the setting of immune dysregulation, contributing to a defective humoral immune response to the vaccine. Similar results have been described by Bergman et al. that propose CD21^low^ B-cell frequency as a marker of poor immunity to Pfizer-BioNTech mRNA anti-COVID vaccine in a cohort of patients with common variable immunodeficiency (CVID) [27].

Similar antigen-specific CD21^low^ B cells have been found to have increased after influenza vaccination, which might in part also contribute to the increase of total CD21^low^ B cells observed in our cohort of patients, in particular because of TLR7 stimulation provided by the Pfizer-BioNTech mRNA-based vaccine.

In these challenging times, there is an urgent need to develop broad-scale population-based tests to improve COVID-19 prevention and diagnosis, as well as to control outcomes and monitor herd immunity [28].

We can suppose that the high percentage of CD21^low^ B cells represents a negative factor of vaccine response because they are exhausted memory B cells, unable to participate in a functional immune response.

Limitations of this study are represented by small sample size, lack of anti-spike IgG serum levels at baseline, and the absence of SSc patients who had undergone immunosuppressive therapy. However, it presents potential utility for monitoring local adaptive immunity. Moreover, the lack of evaluation of B-cell subpopulations after vaccination in HCs is another limitation of this study.

We can conclude that the response to the SARS-CoV-2 mRNA vaccine is normal in SSc patients not undergoing immunosuppressant therapy. The normal number of naive B cells is fundamental for an adequate antibody response to vaccine; conversely, the increased number of CD21^low^ B cells might represent a negative prognostic factor of antibody response. Further larger studies are needed to confirm our preliminary findings.

## Figures and Tables

**Figure 1 jpm-12-01420-f001:**
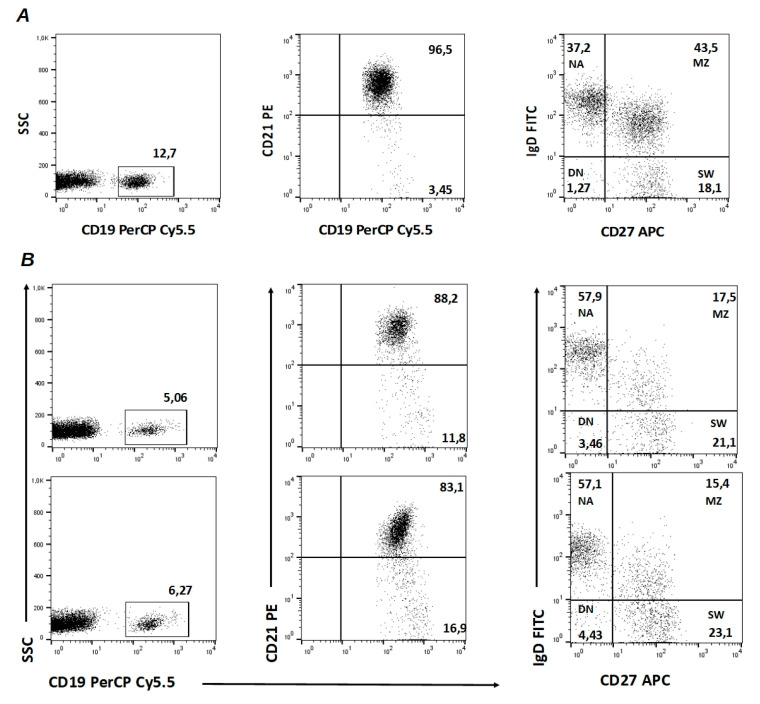
Gating strategy of CD21^low^ B cells and B-cell subpopulations in (**A**) one healthy control (HC) at t1 and in (**B**) one representative systemic sclerosis (SSc) patient at t1 (upper panel) and at t3 (lower panel). NA: IgD^+^ CD27^−^ naive B cells; MZ: IgD^+^ CD27^+^ marginal-zone B cells; SW: IgD^−^ CD27^+^ switched-memory B cells; DN: IgD^−^ CD27^−^ double-negative B cells.

**Figure 2 jpm-12-01420-f002:**
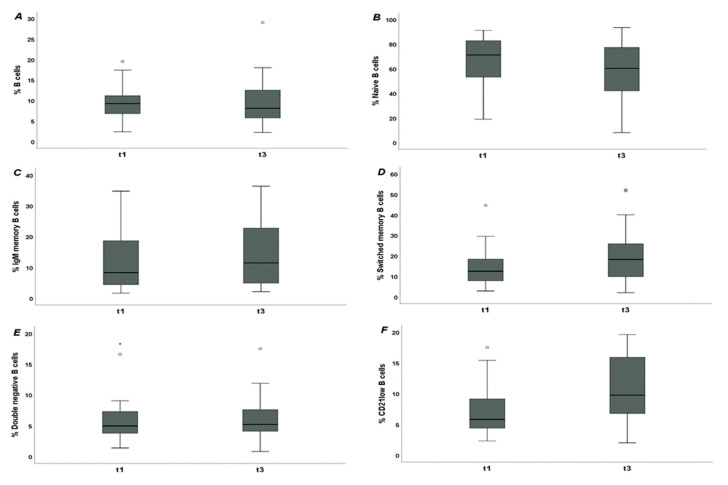
B cells in systemic sclerosis (SSc) patients before (t1) and 3 months after SARS-CoV-2 mRNA vaccine (t3). (**A**): percentage of B cells [8.1% (IQR 5.75–12.5) vs. 7.53% (IQR 5.87–11.5), *p >* 0.05]; (**B**): percentage of naive B cells [60.2% (IQR 41.95–77.25) vs. 70.55% (IQR 47.15–80.2), *p <* 0.01]; (**C**): percentage of IgM-memory B cells [11.45% (IQR 4.9–22.75) vs. 8.57% (IQR 4.62–19.15), *p <* 0.05]; (**D**): percentage of switched-memory B cells [18.25% (IQR 9.85–25.85) vs. 14.15% (IQR 8.5–20.2), *p <* 0.001]; (**E**): percentage of double-negative B cells [5.2% (IQR 4.1–7.6) vs. 5.16% (IQR 3.78–7.76), *p >* 0.05]; (**F**): percentage of CD21^low^ B cells [9.75% (IQR 6.77–15.9) vs. 5.59% (IQR 3.84–9.13), *p <* 0.001].

**Figure 3 jpm-12-01420-f003:**
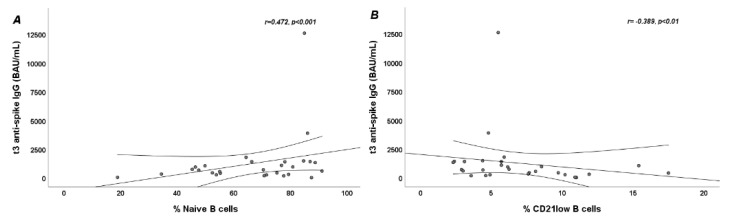
Scatter plots showing linear correlation between B-cell subpopulations and t3 anti-spike IgG serum levels in systemic sclerosis (SSc) patients. (**A**): Positive linear correlation between percentage of naive B cells and t3 anti-spike IgG serum levels (*p <* 0.001); (**B**): Negative linear correlation between percentage of CD21^low^ B cells and t3 anti-spike IgG serum levels (*p <* 0.01).

**Table 1 jpm-12-01420-t001:** Demographic and clinical features of 28 systemic sclerosis (SSc) patients and 21 healthy controls (HCs).

Variables	SSc Patients	HCs	*p*
Age, years, median and IQR	51 (35–62)	48 (33–60)	>0.05
Female, *n* (%)	24 (85.7)	19 (90.5)	>0.05
dcSSc, *n* (%)	13 (46.4)	NA	NA
Disease duration, years, median and IQR	14 (10–21)	NA	NA
mRSS, median and IQR	13 (9–21)	NA	NA
SSc-specific autoantibodies			
Anti-topoisomerase I, *n* (%)	13 (46.4)	NA	NA
Anti-centromere, *n* (%)	10 (35.7)	NA	NA
Anti-RNApolymerase III, *n* (%)	1 (3.6)	NA	NA
None, *n* (%)	4 (14.3)	NA	NA
Nailfold capillaroscopic pattern			
Early, *n* (%)	4 (14.3)	NA	NA
Active, *n* (%)	5 (17.9)	NA	NA
Late, *n* (%)	19 (67.9)	NA	NA
DAI, median and IQR	1.38 (0.88–3.84)	NA	NA
DSS, median and IQR	4 (3–6)	NA	NA
DUs’ history, *n* (%)	14 (50)	NA	NA
Active DUs, *n* (%)	3 (10.7)	NA	NA
ILD, *n* (%)	22 (78.6)	NA	NA
PAH, *n* (%)	2 (7.1)	NA	NA

SSc: systemic sclerosis; HCs: healthy controls; dcSSc: diffuse cutaneous systemic sclerosis; mRSS: modified Rodnan skin score; DAI: Disease activity index; DSS: Disease severity scale; DUs: digital ulcers; ILD: interstitial lung disease; PAH: pulmonary arterial hypertension; IQR: interquartile range; NA: not applicable.

**Table 2 jpm-12-01420-t002:** B-cell subset distribution at t1 and serum anti-spike IgG levels in 28 systemic sclerosis (SSc) patients and 21 healthy controls (HCs).

Variables	SSc Patients	HCs	*p*
Anti-spike IgG levels, BAU/mL, median and IQR	635 (285–1135)	358 (177–1021)	>0.05
B cells, %, median and IQR	7.53 (5.87–11.5)	7.6 (6.86–10.2)	>0.05
Naive B cells, %, median and IQR	70.55 (47.15–80.2)	55.4 (44.5–62.7)	<0.05
IgM-memory B cells, %, median and IQR	8.57 (4.62–19.15)	21.3 (15.8–29.8)	<0.001
Switched-memory B cells, %, median and IQR	14.15 (8.5–20.2)	19.5 (16.1–21.9)	<0.05
Double-negative B cells, %, median and IQR	5.16 (3.78–7.76)	3.96 (3.03–5.1)	<0.05
CD21^low^ B cells, %, median and IQR	5.59 (3.84–9.13)	3.65 (3.11–4.69)	<0.05
Increased naive B cells, *n* (%)	8 (28.6)	0 (0)	<0.01
Increased IgM-memory B cells, *n* (%)	0 (0)	0 (0)	NA
Increased switched-memory B cells, *n* (%)	1 (3.6)	2 (9.5)	>0.05
Increased double-negative B cells, *n* (%)	10 (35.7)	0 (0)	<0.001
Increased CD21^low^ B cells, *n* (%)	6 (21.4)	0 (0)	<0.01

SSc: systemic sclerosis; HCs: healthy controls; IQR: interquartile range; NA: not applicable. The cutoff values for increased B-cell subset distribution were chosen as media +2 standard deviation of the B-cell subset percentage of HCs as follows: increased naive B cells ≥ 80.4%, increased IgM-memory B cells ≥ 43.8%, increased switched-memory B cells ≥ 36.2%, increased double-negative B cells ≥ 6.5%, and increased CD21^low^ B cells ≥ 9%.

**Table 3 jpm-12-01420-t003:** B-cell subset distribution at t1 and t3 in 28 systemic sclerosis (SSc) patients.

Variables	t1	t3	*p*
B cells, %, median and IQR	7.53 (5.87–11.5)	8.1 (5.75–12.5)	>0.05
Naive B cells, %, median and IQR	70.55 (47.15–80.2)	60.2 (41.95–77.25)	<0.05
IgM-memory B cells, %, median and IQR	8.57 (4.62–19.15)	11.45 (4.9–22.75)	<0.001
Switched-memory B cells, %, median and IQR	14.15 (8.5–20.2)	18.25 (9.85–25.85)	<0.05
Double-negative B cells, %, median and IQR	5.16 (3.78–7.76)	5.2 (4.1–7.6)	>0.05
CD21^low^ B cells, %, median and IQR	5.59 (3.84–9.13)	9.75 (6.77–15.9)	<0.05

SSc: systemic sclerosis; IQR: interquartile range.

**Table 4 jpm-12-01420-t004:** B-cell subset distribution at t1 and t3 and serum anti-spike IgG levels in 28 systemic sclerosis (SSc) patients stratified according to antibody positivity and interstitial lung disease (ILD).

		Anti-Topoisomerase I and Anti-RNA Polymerase III (*n* = 14)	Anti-Centromere and ANA (*n =* 14)	*p*	ILD (*n =* 22)	No ILD (*n =* 6)	*p*
Anti-spike IgG levels, BAU/mL, median and IQR		575 (266–946)	1003 (335–1400)	>0.05	575 (266–1050)	1245 (706–1420)	>0.05
t1	Naive B cells, %, median and IQR	68.35 (52.4–75.2)	77.8 (55.2–84.6)	>0.05	71.1 (52.4–84.6)	73.6 (64.3–78)	>0.05
	IgM-memory B cells, %, median and IQR	11.85 (4.44–18.9)	6.61 (4.28–14.7)	>0.05	8.29 (4.35–18.9)	9.64 (6.44–14.7)	>0.05
	Switched-memory B cells, %, median and IQR	15.4 (9.07–18.7)	12.25 (7.46–17.5)	>0.05	13.25 (7.46–18.7)	12.45 (8.78–14.3)	>0.05
	Double-negative B cells, %, median and IQR	5.72 (3.67–8.51)	4.96 (3.88–6.43)	>0.05	5.16 (3.06–7.98)	4.72 (4.25–6.43)	>0.05
	CD21^low^ B cells, %, median and IQR	5.19 (4.4–8.12)	6.07 (4.38–11)	>0.05	6.2 (4.4–10.2)	5.7 (2.89–5.9)	>0.05
t3	Naive B cells, %, median and IQR	55.35 (44.8–77)	64.35 (41.2–77.5)	>0.05	66.95 (44.8–77.5)	44.6 (33–61.5)	>0.05
	IgM-memory B cells, %, median and IQR	14.15 (4.7–26.6)	10.8 (5.1–19.4)	>0.05	11.3 (4.6–20.6)	16.9 (10.4–29)	>0.05
	Switched-memory B cells, %, median and IQR	18.45 (9.7–25.8)	17.7 (11.4–32)	>0.05	17 (9.7–25.8)	27.65 (18.4–32.8)	>0.05
	Double-negative B cells, %, median and IQR	4.3 (3.2–8.7)	5.85 (5–6.8)	>0.05	5 (4–6.8)	6.4 (5.3–9.7)	>0.05
	CD21^low^ B cells, %, median and IQR	9.65 (6.35–13.8)	10.95 (7.18–16)	>0.05	9.65 (6.35–16)	10.55 (8–13.7)	>0.05

SSc: systemic sclerosis; ILD: interstitial lung disease; IQR: interquartile range.

## Data Availability

The original data presented in the study are included in the manuscript. Further inquires can be directed to the corresponding author.

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
