# Peer review of "The Effect of SARS-CoV-2 Vaccination on B-Cell Phenotype in Systemic Sclerosis Patients"

_jpm, 2022, doi:10.3390/jpm12091420_

Round 1
Reviewer 1 Report
In their short report, the Authors assessed the response to vaccination against SARS-CoV-2 and changes in the B-cell subpopulation in patients with systemic sclerosis and healthy volunteers.
Due to some ambiguities, the correct analysis and interpretation of the results requires some clarifications.
- Have patients received any immunosuppressive treatment other than MMF and rituximab in the last 6 months?
- What was the clinical characteristics of the control group?
- Changes in the individual B cell subpopulations before and after vaccination should also be included in the table for easier interpretation of the changes.
- What were the values of the B cell subpopulation in healthy volunteers after vaccination? Was the direction of the changes similar to that in patients with systemic sclerosis or different?
- Was the correlation between the titer of antibodies and B lymphocytes similar in the control group to the results obtained in patients with systemic sclerosis?
- In the discussion, the authors cite that 'Increased prevalence of non response to vaccine was observed in patients with ASD related interstitial lung disease". Were ILD patients different in response or B cell populations from the non-ILD patients in the study group?
- The discussion is superficial, there is no reference to the existing literature examining the relationship between vaccines and B-cell subpopulations, e.g.
Morales-Núñez JJ, García-Chagollán M, Muñoz-Valle JF, Díaz-Pérez SA, Torres-Hernández PC, Rodríguez-Reyes SC, Santoscoy-Ascencio G, Sierra García de Quevedo JJ, Hernández-Bello J. Differences in B-Cell Immunophenotypes and Neutralizing Antibodies Against SARS-CoV-2 After Administration of BNT162b2 (Pfizer-BioNTech) Vaccine in Individuals with and without Prior COVID-19 - A Prospective Cohort Study. J Inflamm Res. 2022 Aug 4;15:4449-4466. doi: 10.2147/JIR.S374304.
- Is the response of scleroderma patients without immunosuppression similar to that of healthy volunteers based on the available publications?
Author Response
I have attached PDF files

Author Response
I have attached PDF file.

Round 2
Reviewer 1 Report
The authors improve the work according to comments from reviewers.
I have no further comments.